# Biochemical Basis of Anti-Cancer-Effects of Phloretin—A Natural Dihydrochalcone

**DOI:** 10.3390/molecules24020278

**Published:** 2019-01-13

**Authors:** Bu Young Choi

**Affiliations:** Department of Pharmaceutical Science & Engineering, Seowon University, Cheongju, Chungbuk 361-742, Korea; bychoi@seowon.ac.kr; Tel.: +82-043-299-8411

**Keywords:** phloretin, cell proliferation, inflammation, apoptosis, glucose uptake, migration

## Abstract

Apple is a rich source of bioactive phytochemicals that help improve health by preventing and/or curing many disease processes, including cancer. One of the apple polyphenols is phloretin [2′,4′,6′-Trihydroxy-3-(4-hydroxyphenyl)-propiophenone], which has been widely investigated for its antioxidant, anti-inflammatory and anti-cancer activities in a wide array of preclinical studies. The efficacy of phloretin in suppressing xenograft tumor growth in athymic nude mice implanted with a variety of human cancer cells, and the ability of the compound to interfere with cancer cells signaling, have made it a promising candidate for anti-cancer drug development. Mechanistically, phloretin has been reported to arrest the growth of tumor cells by blocking cyclins and cyclin-dependent kinases and induce apoptosis by activating mitochondria-mediated cell death. The blockade of the glycolytic pathway via downregulation of GLUT2 mRNA and proteins, and the inhibition of tumor cells migration, also corroborates the anti-cancer effects of phloretin. This review sheds light on the molecular targets of phloretin as a potential anti-cancer and anti-inflammatory natural agent.

## 1. Introduction

Cancer, a heterogenous disease process, still remains a major challenge for human health. Despite the development of a wide variety of anti-cancer therapies such as alkylating agents, various kinase inhibitors, hormone modulators, and very recently introduced immune checkpoint inhibitors, the incidence of and mortality from cancer is still increasing throughout the world. Based on the current trend, unless effective therapeutic interventions are available, a two-fold increase of cancer-related deaths is expected in the next 50 years [1]. A major hurdle to developing anti-cancer therapy is the heterogenous nature of the disease, which often leads to chemotherapy failure or increased resistance to the therapy, thereby allowing recurrence of cancer [2]. Although the neoplastic transformation of cells begins through genetic alterations, such as activation of oncogenes and/or suppression of tumor suppressor genes via chromosomal abrasions, DNA damage, and dysregulation of epigenetic signaling, it has been well documented that various environmental and lifestyle related factors are the major triggers for tumor-inducing genetic and epigenetic changes. Major lifestyle factors that contribute to carcinogenesis include, but are not limited to, consumption of carcinogens from diet, exposure to solar radiation, intake of alcohol, smoking, and lack of physical exercise. Many of these are modifiable factors and thus, cancer can be prevented by adopting an appropriate lifestyle. About half a century ago, the concept cancer prevention was proposed by Michael B. Sporn, who first coined the term “chemoprevention”, which refers to the inhibition or reversal of tumorigenic process by non-toxic chemicals [3].

Over the last several decades, the concept of cancer chemoprevention has been matured through extensive research-based proof-of-principle. Although chemoprevention is regarded as intervening with the pathophysiologic course of tumor development—the initiation, promotion and progression stages [4,5]—based on the clinical perspective, chemoprevention is also classified as either primary, secondary or tertiary prevention of cancer. Any approach directed to prevention of cancer among relatively healthy and at risk groups within a population is termed as primary chemoprevention, while halting the premalignant lesions to complete neoplasia is termed as secondary chemoprevention. The approach of preventing the recurrence of cancer after successful therapy is conceptually termed as tertiary chemoprevention [1,6,7]. Mechanistically, the course of neoplastic transformation of cells can be prevented by intervening with various biochemical processes that become abnormal as a result of DNA damage, and oxidative modifications of proteins and lipids upon exposure to potential carcinogens. These biochemical processes include, but are not limited to, alterations in the activities of enzymes involved in carcinogen metabolism and detoxification, inappropriate amplification of intracellular signaling pathways regulating cell proliferation and apoptosis, tumor angiogenesis, invasion, metastasis, and dysregulation of host immune functions. The key driving forces for all the cancer-related biochemical changes are the persistent oxidative and/or inflammatory stresses imposed on cells or tissues, leading to cellular transformations. Thus, curbing oxidative damage to cellular macromolecules and mitigating inflammatory tissue damage can help prevent tumor development and progression.

A wide array of preclinical and human intervention studies have revealed the potential of various antioxidant and anti-inflammatory phytochemicals to prevent carcinogenesis [2,8]. In fact, many elements of a vegetarian diet, especially fruits and vegetables, contain numerous phytochemicals which can protect cells or tissues from noxious oxidative and inflammation-mediated tissue damage, and hence can prevent the development of tumors. Extensive research has been carried out to explore anti-cancer phytochemicals from fruits and vegetables such as turmeric, broccoli, green tea, grapes, berries, apples, pomegranate, etc. Among those dietary sources, apple is a source of many antioxidant and anti-inflammatory agents. There is a proverb saying that having an apple a day can keep the doctor away. One of the polyphenols present in apple is phloretin [2′,4′,6′-Trihydroxy-3-(4-hydroxyphenyl)-propiophenone], which has received much attention for anti-cancer drug discovery. Being a chalcone compound and having unsaturation and phenolic hydroxyl moieties, phloretin has shown the potential to modify various protein functions, leading to reversal of abnormal signaling and cellular transformation. This review sheds light on the potential of developing phloretin as an anticancer therapy with special focus on its underlying molecular mechanisms.

## 2. Anti-Inflammatory and Anticancer Effects of Phloretin—Evidence from In Vivo Studies

In addition to its nutritive values, apple (*Malus* spp., Rosaceae) can be used for the prevention and treatment of various diseases. The anti-cancer activity of apple products has been evaluated in a large number of epidemiological and laboratory-based studies [9,10]. For example, a significant reduction in the number and size of polyp in the colon and intestine of *adenomatous polyposis* (*APC*)^min+/+^ mice has been observed after allowing these animals to drink a polyphenolic extract obtained from apple for 12 weeks [11]. This popular fruit is a rich source of various bioactive chemicals, such as chalcones, flavonoids, procyanidins and terpenoids [9]. One of the bioactive constituents of apple is phloretin [12], which has been widely investigated for its anti-cancer activities. In a pioneering study, Tanaka et al. [13] first demonstrated that phloretin inhibited the neoplastic transformation of BALB/3T3 cells upon exposure to a prototypic tumor promoting agent 12-*O*-tetradecanoyl phorbol-13 acetate (TPA). Subsequent studies have reported that phloretin exhibits antioxidative, anti-inflammatory, antiproliferative and apoptosis inducing properties. In addition, when used alone or in combination with conventional chemotherapy, phloretin can suppress the in vivo growth of xenograft tumors in nude mice [12,14,15,16]. For example, phloretin significantly reduced the volume and weight of xenograft tumors in severe combined immunodeficiency (SCID) mice inoculated with human hepatocellular carcinoma (HepG2) cells [17]. Phloretin elicited inhibitory effects on the growth of human lung adenocarcinoma (A549) cell xenograft tumors in nude mice [18] and enhanced the growth inhibitory effect of paclitaxel in HepG2 cell xenograft tumors in SCID mice [15]. Further evidence of in vivo anti-tumor effects of phloretin were reported by Shin et al. [19], who demonstrated that topical application of phloretin significantly reduced the multiplicity of papillomas in 7,12,-dimethylbenz(a)anthracene (DMBA)-initiated and TPA-promoted mouse skin. Oral administration of phloretin also attenuated DMBA-induced buccal pouch carcinogenesis in male Syrian golden hamsters [20]. In healthy human volunteers, topical application of an antioxidant formulation containing vitamin C, ferrulic acid and phloretin suppressed ultra violet (UV) radiation-induced sunburn, thymidine dimer formation, and matrix metalloproteinase-9 (MMP-9) expression in skins [21]. Since exposure to UV radiation is a major cause of skin cancer, this study suggests the potential of phloretin to inhibit UV-induced skin carcinogenesis.

The link between chronic inflammation and cancer has been well documented. Amelioration of colitis, a condition characterized by persistent colonic mucosal inflammation, often progresses to colorectal cancer [22]. Management of colitis with anti-inflammatory therapy reduces the risk colorectal cancer [23]. Administration of phloretin significantly ameliorated trinitrobenzene sulfonic acid (TNBS)-induced colon inflammation and the loss of body weight in rats [24], suggesting the potential of this compound to prevent colorectal carcinogenesis. This has been supported by a recent study demonstrating the effectiveness of phloretin in suppressing the growth of human colorectal cancer (COLO 205) tumor xenografts in Balb/c nude mice [25]. When given intraperitoneally, phloretin inhibited ovalbumin-induced airway inflammation in Balb/C mice [26]. Moreover, intraperitoneal administration of phloretin diminished cigarette smoke (CS)-induced secretion of mucins, infiltration of inflammatory cells and the release of inflammatory cytokines in mouse lungs, and attenuated CS extract-induced expression of MUC5AC and IL-1β in NCI-H292 bronchial epithelial cells, partly by blocking the activation of EGFR, ERK and p38 mitogen-activated protein (MAP) kinase [27]. Thus, it would be interesting to examine whether phloretin can suppress CS-induced lung carcinogenesis. Phloretin was reported to inhibit the xenograft tumor growth of human lung cancer (A549) cells [18] and human breast cancer (MDA-MB-231) cells [28] in Balb/c nude mice.

## 3. The Biochemistry behind the Anti-Inflammatory and Anti-Cancer Effects of Phloretin

The biochemical mechanisms of anti-inflammatory and anti-cancer effects of phloretin have been widely investigated (Table 1). Phloretin has been shown to scavenge peroxynitrite radicals and inhibit lipid peroxidation, largely due to the presence of 2,6-dihydroxyacetone moiety as the 130 pharmacophore [29]. In another study, Nakamura et al. [12] demonstrated that the presence of a hydroxyl group at 2′-position of all dihydrochalcones, including phloretin, is an essential pharmacophore for the radical scavenging and lipid peroxidation activities. The bond dissociation enthalpies (BDEs) for hydroxyl (-OH) moieties (an indicator of antioxidant activity) of phloretin has recently been examined in a computational modeling study. The authors demonstrated that compared to well-known antioxidants, the BDEs among the four OH groups for phloretin were found to be the lowest, indicating the antioxidant potential of the compound [30]. Phloretin prevented oxidative DNA damage by restoring cellular glutathione (GSH) levels in human colon cancer (Caco-2 and HT-29) cells [31]. Likewise, the elevated cellular GSH level in phloretin-treated rat hepatocytes has been attributed to the induction of γ-glutamyl cysteine ligase (GCL), a rate limiting enzyme in GSH synthesis, via activation of a redox-regulated transcription factor, nuclear factor-erythroid related factor-2 (Nrf2) signaling [32]. These authors also demonstrated that phloretin induced the expression of another Nrf-2-regulated antioxidant enzyme, hemeoxygenase-1 (HO-1), in rat hepatocytes, via the activation of extracellular signal regulated kinase (ERK). By virtue of its antioxidative properties, phloretin attenuates the activation of exogenously exposed carcinogens as well as endogenous accumulation of damaged DNA or proteins, and inhibits the aberrant activation of various kinases and transcription factors involved in inflammatory signaling pathways. The following section will focus on detailing the molecular targets of phloretin as an anti-cancer agent (Figure 1).

### 3.1. Inhibitory Effects of Phloretin on Inflammatory Markers

While inflammatory tissue damage is an initial trigger for tumor development, persistent intra-tumoral inflammation contributes to the invasion and metastasis of tumors. The key mediators of inflammation involved in neoplastic transformation of cells include, but are not limited to, inducible nitric oxide synthase (iNOS) and cyclooxygenase-2 (COX-2), and the cellular products catalyzed these enzymes, such as nitric oxide (NO) and prostaglandins (PGs) [22]. Studies have shown that mice harboring transgenic overexpression of *cox-2* in the stomach [44] and skin [45] are highly prone to develop tumors in these organs, whereas genetic ablation of *cox-2* protects mice against gastro-intestinal [46] and skin carcinogenesis [47]. The elevated levels of iNOS and its catabolic product NO have promoted dextran sulfate sodium-induced polyp formation in the colon of *APC*^min+^ mice as compared to wild type mice [48]. The role of iNOS in skin papillomas has been evident from the inhibition of chemically induced mouse skin tumor development upon treatment with aminoguanidine, which is an iNOS inhibitor [49]. A large number of cytokines, for example, interleukins (IL) and tumor necrosis factor-α (TNFα), and chemokines also act as inflammatory mediators and participate in the tumorigenic process. The burst of inflammatory mediators requires inappropriate amplification of intracellular signaling cascades comprising various kinases and transcription factors. Aberrant activation of MAP kinases, Janus-activated kinase (JAK), and protein kinase B (PKB)/Akt have been implicated in precipitating inflammation and cancer. These upstream kinases transmit activating signals to a variety of redox-sensitive transcription factors, such as nuclear factor-kappaB (NF-κB), activator protein-1 (AP-1), signal transducer and activator of transcription (STAT), which leads to transactivation of genes encoding proteins involved in inflammation and oncogenesis [8,50].

Phloretin exerts anti-cancer effects by curbing inflammatory responses. For instance, phloretin markedly diminished the expression of pro-inflammatory genes by repressing the activation of NF-κB-, IL-8-, and STAT1-dependent signal transduction in human colon cancer (DLD1) cells and other immune cells (T84, MonoMac6, Jurkat) in a concentration-dependent manner [51]. The compound also inhibited TNFα-induced inflammatory responses in human colonic epithelial cells [24]. Shin et al. [19] reported that topical application of phloretin reduced phorbol ester-induced COX-2 expression and skin inflammation in mice by blocking the activation of NF-κB and ERK-mediated signaling. Moreover, pretreatment with phloretin diminished the production of various inflammatory markers, such as NO, PGE_2_, IL-6, and TNFα, and attenuated the expression of iNOS and COX-2 in lipopolysaccharide (LPS)-stimulated murine macrophage RAW264.7 cells. According to this study, phloretin blocked nuclear localization of the p65 protein, a component of NF-κB, and negated the phosphorylation in MAP kinases [16]. Phloretin reduced the expression of COX-2 and intracellular adhesion molecule -1 (ICAM-1), and the production of IL-6 in human lung epithelial (A549) cells stimulated with IL-1β by blocking the activation of NF-κB via downregulation of Akt and MAP kinases phosphorylation [35]. Likewise, the secretion of various cytokines and chemokines, such as IL-6, IL-8, and monocyte chemoattractant protein-1(MCP1), and the reduced expression of ICAM-1 in TNF-α-stimulated HaCaT keratinocytes by phloretin, have been attributed to the inactivation of NF-κB and MAP kinases [36]. Huang et al. recently reported that treatment with phloretin attenuated the gene expression of a variety of inflammatory markers, such as COX-2, iNOS, CCL5, MCP1, and ICAM-1 in LPS-treated mouse lung tissue [22,34,36,52,53].

### 3.2. Modulating Biotransformation of Putative Carcinogens

Cellular transformation is often led by exposure to potential carcinogenic stimuli. While many environmental or dietary factors are non-toxic, some can be turned into potential genotoxic agents once metabolically activated. A rational approach for preventing cells from potential toxic effects of carcinogens is either to block metabolic activation of the apparently nontoxic chemicals or to enhance cellular detoxification pathways to eliminate the noxious chemical entities. A wide array of cytochrome (CYP) P450 enzymes carry out the bioactivation of pro-carcinogens, whereas a series of detoxification enzymes, such as glutathione-*S*-transferase (GST), glucuronyl transferase, sulfotransferase, etc., help in the elimination of metabolically active carcinogens or their reactive intermediates. Phloretin inhibited the biotransformation of aflatoxin B1 (AFB1) to generate AFB1-8,9-epoxide (AFBO), an active metabolite of AFB1, by blocking the activities of CYP1A2 and CYP3A4 enzymes. Moreover, phloretin induced the activity of GST and increased the expression of GSTA3, GSTA4, GSTM1, GSTP1 and GSTT1 via the activation of Nrf2 in alpha mouse liver 12 (AML12) cells [54]. Since metabolically activated AFB1 acts as a potential hepatocarcinogen and causes liver cancer, phloretin may prevent chemically induced liver cancer. Phloretin increased the expression of Nrf2, a redox-regulated transcription factor involved in transactivation of antioxidant and detoxification genes, and upregulated the expression of heme oxygenase-1 (HO-1) in TNFα-stimulated HaCaT keratinocytes [35] and LPS-treated mouse lung tissue [34], suggesting the potential of phloretin to protect against oxidative tissue damage in these organs.

### 3.3. Antiproliferative and Apoptosis Inducing Effects of Phloretin

There is accumulating evidence suggesting that phloretin arrests the proliferation of tumor cells and induces apoptosis in various human cancer cells, including those of the skin, colon, breast and prostate in culture. The underlying mechanisms of the antiproliferative effects of phloretin appears to be the interference with cell cycle progression via modulation of cyclins and induction of mitochondria-dependent programmed cell death. Park et al. [14] reported that the induction of apoptosis in human colon cancer (HT-29) cells upon treatment with phloretin was associated with increased expression of Bax and release of cytochrome c and Smac/DIABLO in the cytosol, thereby resulting in the cleavage of caspase-8, -9, -7, and -3 and poly(ADP-ribose) polymerase (PARP) [14]. A recent study demonstrated that phloretin induced apoptosis selectively in human gastric cancer cell lines (MGC80-3, BGC-823, SGC-7901, SNU-1, SNU-5, RF-1 and AGS), with IC_50_ values ranging from 8 to 32 µM without affecting the growth of normal gastric epithelial cells [39]. Incubation of AGS cells with phloretin reduced the colony formation in a concentration-dependent manner by arresting cell cycle at the G2/M phase. Moreover, the induction of apoptosis in AGS cells by phloretin was associated with the increase in Bax expression, inhibition of Bcl-2 levels and decrease in the phosphorylation of c-Jun N-terminal kinase (JNK) and p38 MAP kinase [39]. In contrast, the activation of JNK and p38 MAP kinase has been attributed to phloretin-induced caspase-3 cleavage in H-*ras*-transformed human mammary epithelial cells (H-*ras*-MCF-10A) [55]. These authors also reported that phloretin attenuated the proliferation of H-*ras*-MCF-10A cells in a concentration-dependent manner and induced apoptosis via upregulation of p53 and Bax and the cleavage of PARP [55]. Although the induction of apoptosis in B16 melanoma cells by phloretin was associated with the elevated expression of Bax and caspase activation, the compound did not affect the expression of p53 or that of anti-apoptotic proteins Bcl-2 or Bcl-xl [56]. However, according to a recent study phloretin inhibited the expression of Bcl-2 and X-linked inhibitor of apoptosis (XIAP), and activated p53 as the mechanism of apoptosis induction in human esophageal cancer (EC-109) cells [40]. Phloretin arrested cell cycle at the G0/G1 phase and reduced proliferation of human breast cancer (MDA-MB-231) cells in a p53 mutant-dependent manner as evidenced by pre-incubating cells with a p53-specific dominant-negative expression vector [28]. Thus, the effects of phloretin on MAP kinase activation, Bcl-2 expression and p53 induction may be a cell-type specific phenomena.

Kobori et al. reported that phloretin caused apoptosis in human leukemia (HL60) cells, which was associated with decreased protein kinase C (PKC) activity [57]. Various chalcone compounds have been shown to prevent prostate cancer [58]. Phloretin, being a dihydrochalcone, significantly enhanced TNFα-related apoptosis-inducing ligand (TRAIL)-induced apoptosis and cytotoxicity in human prostate cancer (LNCaP) cells [58].

Phloretin attenuated growth and induced apoptosis in a number of lung cancer cells in culture. Treatment of human lung cancer (A549, Calu-1, H838 and H520) cells with phloretin resulted in reduced expression of anti-apoptotic protein Bcl-2 and elevation in the protein expression of cleaved-caspase-3 and -9 [41]. The induction of apoptosis in A549 cells by phloretin was associated with increased levels of Bax, cleavage of caspase-3 and -9, and PARP, and decreased Bcl-2 expression. Mechanistically, phloretin-induced caspase activation was mediated through the induction of p38 MAP kinase and JNK1/2 phosphorylation, as inhibition of these kinases by using specific inhibitors significantly abolished the phloretin-induced caspase activation [18].

The anti-proliferative effects of phloretin are also mediated through the modulation of cyclins and cyclin-dependent kinases (CDKs). According to a recent study, phloretin induced cell cycle arrest at the G0/G1 phase in human glioblastoma cells, at least in part, by increasing the expression of p27 and dampening that of CDK-2, -4, and -6, and cyclin-D, and -E. In addition, the compound suppressed signaling through the PI3K/AKT/mTOR cascades, resulting in reduced cell proliferation. The study also demonstrated that phloretin triggered the mitochondria-mediated cell death via generation of ROS, up-regulation of Bax, Bak and c-PARP, and the inhibition of Bcl-2 [59].

### 3.4. Blockade of Tumor Cell Migration and Invasion

One of the pathophysiologic features of carcinogenesis is the spreading of localized tumors to distant tissue sites through migration and invasion [60]. Tumor cell migration involves epithelial-mesenchymal transition (EMT) followed by matrix degradation to invade through stromal tissue [61]. While upregulation of N-cadherin and vimentin are known markers of EMT, matrix metalloproteinases (MMPs) are essential molecules that cause matrix degradation [62]. Several studies have demonstrated the ability of phloretin to reduce the migratory potential of various tumors cells, thus blocking tumor progression. Incubation with phloretin diminished migration of human lung epithelial cancer (A549) cells in a concentration-dependent manner, which was associated with downregulation of NF-κB and MMP-9 expression [18]. Likewise, the inhibition of migration and invasion of non-small cell lung cancer (H838 and H520) cells by phloretin treatment was mediated through suppression of MMP-2 and MMP-9 at both gene and protein levels [41]. According to Wu et al., phloretin treatment diminished the expression of paxillin and α-smooth muscle actin (α-SMA), and decreased the phosphorylation of focal adhesion kinase and Src kinase, thereby slowing down the migration of human breast cancer MDA-MB-231 cells. Although phloretin had no remarkable effect on the expression of E-cadherin in MDA-MB-231 cells, the study showed that the compound upregulated E-cadherin and inhibited N-cadherin and vimentin expression in MDA-MB-231 cells xenograft tumors in nude mice. While a decrease in vimentin level is in agreement, the effect of the compound on cadherin expression failed to support its antimigratory effect [28]. Further studies are required to confirm the effects of phloretin on cadherin levels and its impact on tumor cell migration. Phloretin also attenuated the invasion of human gastric cancer (AGS) cells by blocking the phosphorylation of MAP kinases [39].

### 3.5. Inhibition of Glucose Uptake by Phloretin

Enhanced glycolysis, a feature of many rapidly growing solid tumors, is associated with elevated expression of glucose transporter proteins (GLUT) and other glycolytic enzymes in neoplastic cells. Treatment with phloretin sensitized human colon cancer (SW620) and leukemia (K562) cells to daunorubicin by reducing the glucose uptake by these cells [63]. Lin et al. [25] reported that elevated levels of GLUT2 mRNA were detected in human colon cancer (COLO 205 and HT29) cells as well as in colorectal cancer tissues. Treatment with phloretin attenuated GLUT2 mRNA and protein expression by blocking hepatocyte nuclear factor 6 (HNF6), which is a transcription factor regulating GLUT2 expression. When compared to normal hepatocytes, the levels of GLUT2 mRNA were found to be five times greater in human hepatoma (HepG2) cells. Phloretin re-sensitized these cells to paclitaxel and induced apoptosis via caspase activation by attenuating GLUT2 expression. Thus, the growth of HepG2 cells xenograft tumors in nude mice markedly diminished by co-treatment with phloretin and paclitaxel, as compared to treatment with paclitaxel alone [15]. Another study has shown that GLUT2 expression is elevated in HepG2 cells and that the siRNA-mediated silencing of GLUT2 induced apoptosis in these cells [17]. While co-treatment of these cells with glucose uptake inhibitor cytochalasin B enhanced phloretin-induced apoptosis, the effect of phloretin was reversed by pretreatment with glucose. These findings suggest that induction of apoptosis in HepG2 cells by phloretin is mediated through inhibition of GLUT2 expression. Likewise, phloretin inhibited the proliferation of rat mammary adenocarcinoma and Fischer bladder cell carcinoma cells and arrested the growth of xenograft tumors of these cells in mice by blocking glucose transmembrane transport [64].

### 3.6. Phloretin as a Potential Cancer Immunotherapy Agent

The emergence of immunotherapies in recent years is considered a major breakthrough in the development of cancer therapeutics. Reinvigoration of exhausted T cells by blocking immune checkpoint markers, such as cytotoxic T cells antigen-4 (CTLA4), programmed cell death 1 (PD-1) and PD-1 ligand (PDL-1), have recently received Food and Drug Administration approval for the treatment of melanoma, renal cell carcinoma, and non-small cells lung cancer [65]. Although substantial progress has been made in developing cancer immunotherapies, the potential of natural compounds in boosting antitumor immunity is yet to be investigated. A recent study by Zhu et al. [66] reported that phloretin can enhance the tumoricidal effect of γδ T cells on human colon cancer (SW-1116) cells, possibly by stimulating the proliferation of γδ T cells. In another study, phloretin potentiated the anti-cancer effects of intratumorally administered recombinant heat shock protein-70 (HSP70) in B16 mouse melanoma cells, through enhancement of tumor cell sensitivity to cytotoxic lymphocytes by 16%–18% as compared to a treatment with HSP-70 alone. The combined treatment with phloretin and recombinant HSP-70 resulted in B16 melanoma cells xenograft tumor growth and increased the life span of tumor-bearing mice through the activation of innate and adaptive immunity. Authors also demonstrated that rHSP70 was more active in induction of CD8+ cell function and interferon-γ (γIFN) production, but phloretin induced CD56+ cell response [42].

### 3.7. Phloretin Alleviates Chemotherapy Resistance

A major challenge to the clinical success of curing cancer is the frequent incidence of therapy failure, largely due to the acquisition of resistance by tumor cells to ongoing chemotherapy [67,68]. The biochemical mechanism underlying the development of chemoresistance is the ability of tumor cells to enhance drug efflux, thereby reducing the intracellular drug concentration to a sub-therapeutic level [69]. The overexpression of multidrug resistance proteins (MRP or MDR), for example P-glycoprotein (P-gp), on the cancer cell membrane often facilitates drug efflux and helps tumors to gain chemotherapy resistance [70]. Thus, the blockade of MDR-efflux proteins may help promote the effectiveness of cancer chemotherapy. Co-administration of a wide variety of dietary phytochemicals are shown to sensitize chemoresistant tumor cells in preclinical studies. Nguyen et al. [71] demonstrated that the accumulation of daunomycin and vinblastine in human pancreatic carcinoma Panc-1 cells was significantly increased when the cells were co-treated with phloretin and the effect was mediated through the inhibition of MRP1-mediated drug transport. According to another study, treatment with phloretin reduced the P-gp activity in human MDR1 gene-transfected mouse lymphoma cells (L1210) and human breast cancer cells MDA-MB-231 expressing the MRP1 pump (HTB26) [72]. Moreover, phloretin facilitated cisplatin-induced downregulation of Bcl-2, MMP-2 and -9, and activation of caspase cleavage in NSCLC cells [41], while the compound inhibited cisplatin-induced apoptosis of normal auditory cells [73]. These findings suggest that phloretin not only enhances chemotherapeutic effects of cisplatin, but also prevents the potential side effects of cisplatin in causing auditory damage. Zhou et al. [74] examined the growth inhibitory effect of a combination of atorvastatin, an anti-lipidemic agent, and phloretin on SW620 and HCT116 colon cancer cells. The combination treatment caused a remarkable decrease in cell survival in both cell lines as compared to treatment with an individual compound. The computational analysis of the interaction index between phloretin and atorvastatin appeared to be <1.0, indicating a strong synergistic effect exerted by the compounds. Synergistic anti-cancer effects were revealed by the induction of apoptosis and cell cycle arrest at the G2/M phase, which was associated with reduced expression of cyclin B and elevated expression of phospho-cdc2 and Myt1 [74].

## 4. Pharmacokinetics and Toxicity Profile of Phloretin

Although the anti-cancer activities of phloretin have been investigated in various cell culture experiments in vitro and animal models in vivo, there have been limited studies on its pharmacokinetics and toxicity profile. In one study, the plasma and urine analysis of phloretin metabolites in rats fed a single meal containing 0.157% phloretin (corresponding to the ingestion of 22 mg of phloretin equivalents) revealed that phloretin was present in plasma primarily as glucuronide and sulphate conjugates, but unconjugated phloretin was also detected. The study also showed that phloretin appeared more rapidly in plasma when rats were fed the aglycone as opposed to its glucoside form. Moreover, the return of plasma phloretin concentration to its baseline after 24 h of the meal indicates the rapid elimination of the compound from the body. The total urinary excretion rate of phloretin was estimated to be 8.5 µmol/24 h, and about 10.4% of the ingested dose was recovered in urine after 24 h of the phloretin-containing meal [75]. Ingestion of polyphenol-rich cloudy apple juice by healthy volunteers followed by ileostomy extract analysis showed that most of the orally administered apple polyphenols are absorbed from, or metabolized in, the small intestine. Both phloretin and its glucuronide conjugates were detected in the ileostomy extract from 2 to 4 h, indicating that most of the apple juice polyphenols, including phloretin, are well absorbed in the small intestine [76]. In contrast, translocation of phloretin and its glucosides through ex vivo pig intestinal smooth muscle remained undetected, suggesting poor absorption of phloretin [77]. A recent study demonstrated the protective effect of a low dose (0.2 to 0.4 mmol/kg body weight) of phloretin against acetaminophen (AAP)-induced hepatotoxicity in mice, while the compound given intraperitoneally at a dose of 2.4 mmol/kg alone showed 64% lethality [78]. Thus, further studies are warranted to obtain more precise pharmacokinetics parameters, and lethal and sub-acute toxicity for the therapeutic effectiveness of the compound as an anti-cancer agent.

## 5. Future Perspectives

Despite substantial progress in developing molecular target-based cancer chemotherapeutic agents, including the recently introduced immunotherapies, the clinical success of these therapeutic agents remains limited. The risk of developing resistance, high cost and cancer type-specificity narrow down the utility of these agents for treating millions of cancer patients worldwide. A wide variety of phytochemicals present in our regular foods are shown to prevent or delay the carcinogenesis process. The search for dietary anti-cancer principles and the elucidation of their biochemical mechanisms appears as a rational and active branch of new drug discovery research over this century. The beneficial health effects of plant polyphenols have been well documented. One of the dietary polyphenols is phloretin, which has emerged as a promising anti-cancer agent. While diverse biochemical mechanisms of anti-cancer effects of this apple polyphenol have been explored, exploitation of this molecule for new anti-cancer drug design requires more rigorous studies to address issues like toxicity profile, pharmacokinetics and more elaborate molecular basis of anti-cancer activity.

## Figures and Tables

**Figure 1 molecules-24-00278-f001:**
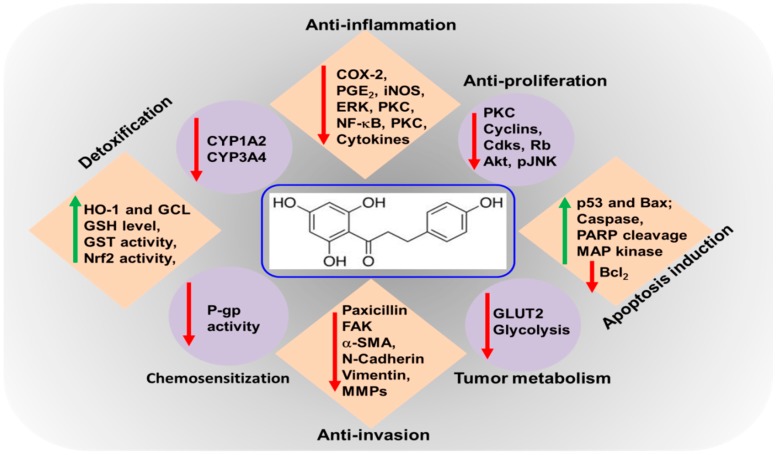
Molecular targets of phloretin as an chemoprevention agent. Anti-inflammation: COX-2 (Cyclooxygenase-2), PGE_2_ (Prostaglandin E_2_), *i*NOS (Nitric oxide synthases), ERK (extracellular-signal-regulated kinases), PKC (Protein kinase C), NF-kB (nuclear factor kappa-light-chain-enhancer of activated B cells). Anti-proliferation: CDKs (Cyclin-dependent kinases), Rb (Retinoblastoma protein), Akt (protein kinase B), p-JNK (Phospho-c-Jun N-terminal kinases), Apoptosis induction: Bax (Bcl-2-associated X protein), PARP (Poly ADP-ribose polymerase), Bcl_2_ (B-cell lymphoma 2). Tumor metabolism: GLUT2 (Glucose transporter 2). Anti-invasion: FAK (focal adhesion kinase), α-SMA (Smooth muscle actin), MMPs (Matrix metalloproteinases). Chemosensitization: P-gp (P-glycoprotein 1). Detoxification: HO-1 (heme oxygenase 1), GCL (Glutamate Cysteine Ligase), GSH (glutathione), GST (Glutathione S-transferases), Nrf2 (Nuclear factor erythroid-related factor-2).

**Table 1 molecules-24-00278-t001:** Anti-inflammatory and anti-cancer activities of phloretin.

Type	Experimental Model	Dose/Concentration	Mechanism of Action	Ref.
Anti-inflammatory	HEK293 cells engineered to either overexpress or deficient in hTLR2	10 or 20 µM	Inhibits the heterodimerization of TLR2/1; reduced the secretion of TNF-α and IL-8	[33]	
Specific pathogen-free male BALB/c mice (6–8 weeks, 22–24 g)	20 mg/kg	Suppressed the mucins secretion, inflammatory cell infiltration and cytokine release in mouse lungs induced by cigarette smoke (CS)	[27]	
OVA-challenged asthmatic mice	5, 10, or 20 mg/kg	Decreased hyperresponsiveness, inflammation, and oxidative responses; reduced ROS, and cytokines production	[26]	
LPS-induced acute lung injury in mice	5 or 20 mg/kg	Suppressed LPS-induced neutrophil infiltration, and reduce the levels of IL-6 and TNF-α in serum and bronchoalveolar lavage fluid; blockade of the NF-κB and MAPK pathways	[34]	
A549 cells	3–100 μM	Inhibited proinflammatory cytokine, COX-2, and ICAM-1 expression; blocked NF-κB and MAPK signaling pathways.	[35]	
TNF-α-stimulated HaCaT human keratinocytes	10, 30, or 100 μM	Decreased the production of IL-6, IL-8, and CCL5; inhibited NF-κB nuclear translocation; suppressed phosphorylation of Akt and MAPK signal.	[36]	
Human THP-1 monocytes	1, 10, or 30 μg/mL	Reduced TNF-α, IL-6 and COX-2 expression	[37]	
Rat basophilic leukemia RBL-2H3 cells	12.5, 25, or 50 μM	Attenuated ROS production, phosphorylation of Akt, ERK1/2, p38 MAP kinase, and JNK	[38]	
LPS-stimulated murine RAW264.7 macrophages	3, 10, 30, or 100 μM	Reduced the levels of NO, PGE2, IL-6, TNF-α, iNOS and COX-2; suppressed nuclear translocation of NF-κB subunit p65 proteins, and decreased phosphorylation of MAPK pathways	[16]	
Anti-cancer	Gastric cancer (AGS) cells	IC_50_ 8 μM	Arrested the cell cycle in G2/M phase and decreased the expression of p-JNK and p-p38 MAP kinase	[39]	
Esophageal cancer EC-109 cell lines	60 µg/mL	Apoptosis increased to 225.6 ± 16.0%; increased p53 activity; increased the level of Bax and declined Bcl-2 levels	[40]	
Non-small cell lung cancer (NSCLC): A549, Calu-1, H838 and H520 cells	25, 50 or 75 μg/mL	Suppressed the expression of Bcl-2; increased the protein expression of cleaved-caspase-3 and -9, and deregulated the expression of MMP-2 and -9 on gene and protein levels	[41]	
Human erythroid leukemia K-562 cells	20 μM	Increased the efficacy of HSP70 penetration; increases anti-tumor activity of HSP70 with phloretin combination	[42]	
A549 human lung cancer cell line, Bel 7402 liver cancer cell line, HepG2 human ileocecal cancer cell line, and HT-29 human colon cancer cell line	0–150 mg/mL	Significant positive anti-cancer activities against several human cancer cell lines, IC_50_: A549 (27 μg/mL), BEL7402 (37 μg/mL, HepG2 (37 μg/mL), HT29 (33 μg/mL)	[43]	
HepG2-xenografted tumor	10 mg/kg phloretin or +1 mg/kg paclitaxel	Reduced tumor growth more than fivefold in the phloretin and paclitaxel-treated mice compared to the paclitaxel only treated mice	[15]

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
