# Peer review of "Biochemical Basis of Anti-Cancer-Effects of Phloretin—A Natural Dihydrochalcone"

_molecules, 2019, doi:10.3390/molecules24020278_

Round 1
Reviewer 1 Report
This is a well written and documented review describing the potential anti-cancer activity of the polyphenol phloretin, a molecule particulary present in apple (Malus spp., Rosaceae). Probably most of the bibliography available in literature about the topic has been cited by the Author.
To be completely exhaustive a paragraph on phloretin pharmacokinetic and toxic profile should be added.
The review focuses on the anti-cancer activity of phloretin rather than on its anti-inflammatory activity that is instead reported as a mechanism at the basis of phloretin anti-cancer activity. Thus, I would suggest to remove “anti-inflammatory” from the title of the article.
This is a well-written report that anyway requires some English editing. Here are reported some examples of corrections needed:
· 37: cancer [2]. Although
· 58: DNA damage, and oxidative modifications
· 83: apple (Malus spp., Rosaceae)
· 132: pharmacophore for the radical scavenging and lipid peroxidation ?
· Figure 1: Nrf2 activity
· 150: Molecular targets of phloretin as a chemopreventive agent
· 153 & 267: Cyclin-dependent kinases should be abbreviated as CDKs not Cdks
· 153: Rb (Retinoblastoma protein)
· 157: P-gp
· Table 1: wrong references are reported all over the table. Almost all are referred to [62], that is not even reported in reference list.
· Table 1: merge the two raws about OVA-challenged asthmatic mice
· Table 1: 12.5, 25, or 50 μM
· Table 1: Arrested the cell cycle in G2/M phase
· Table 1: please improve punctuation in Mechanism of action column
· 168-9: COX-2
· 205-207: correct the sentence
· 205-207: add some reference
· 206: per se
· 210: P450
· 216 and 219: there is no need to repeat what Nrf2 stands for (see raw 141-142)
· 227: and prostate in culture
· 229: mitochondria-dependent programmed cell death
· 231: Smac/DIABLO
· 241: in H-ras-transformed human mammary epithelial cells
· Be consistent in typing proteins or genes (eg. BAX or Bax; ICAM1 or ICAM-1; p38 or P38), chosing the more appropriate form
· Cyclin-dependent kinases should be abbreviated as CDKs not Cdks
· 268: G0/G1
· 276: epithelial-mesenchymal transition (EMT)
· 274-286: Add some reference
· 291: remove a )
· 299: glucose uptake
· 299: reported that elevated
· 302: which is a transcription factor
· 316: Phloretin as a potential cancer immunotherapy agent
· 319: programmed cell death 1 (PD-1)
· 320: PD-L1
· 322: progress has been made
· 336-337: Add some reference
· 336-337: this is not only mechanism as the Author seems to suggest
· 338-9: P-glycoprotein
· 365: to prevent or delay
Author Response
Journal Molecules (ISSN 1420-3049)
Manuscript ID: Molecules-423825
Title : Biochemical basis of anti-cancer effects of Phloretin – A Natural Dihydrochalcone
Reviewer #1
Comments and Suggestions for Authors
This is a well written and documented review describing the potential anti-cancer activity of the polyphenol phloretin, a molecule particulary present in apple (Malus spp., Rosaceae). Probably most of the bibliography available in literature about the topic has been cited by the Author.
Comment 1
To be completely exhaustive a paragraph on phloretin pharmacokinetic and toxic profile should be added.
The review focuses on the anti-cancer activity of phloretin rather than on its anti-inflammatory activity that is instead reported as a mechanism at the basis of phloretin anti-cancer activity. Thus, I would suggest to remove “anti-inflammatory” from the title of the article.
This is a well-written report that anyway requires some English editing.
Response 1
Thank you for your good comments. Manuscript page 10 “ 4. Pharmacokinetics and toxicity profile of phloretin”. The topic was written.
Comment 2: Here are reported some examples of corrections needed:
Response 2
Thank you for your comments. And I have finished editing the comments. The modifications were highlighted in yellow in manuscript.
37: cancer [2]. Although
· 58: DNA damage, and oxidative modifications
· 83: apple (Malus spp., Rosaceae)
· 132: pharmacophore for the radical scavenging and lipid peroxidation ?
· Figure 1: Nrf2 activity
· 150: Molecular targets of phloretin as a chemopreventive agent
· 153 & 267: Cyclin-dependent kinases should be abbreviated as CDKs not Cdks
· 153: Rb (Retinoblastoma protein)
· 157: P-gp
· Table 1: wrong references are reported all over the table. Almost all are referred to [62], that is not even reported in reference list.
· Table 1: merge the two raws about OVA-challenged asthmatic mice
· Table 1: 12.5, 25, or 50 μM
· Table 1: Arrested the cell cycle in G2/M phase
· Table 1: please improve punctuation in Mechanism of action column
· 168-9: COX-2
· 205-207: correct the sentence
· 205-207: add some reference
· 206: per se
· 210: P450
· 216 and 219: there is no need to repeat what Nrf2 stands for (see raw 141-142)
· 227: and prostate in culture
· 229: mitochondria-dependent programmed cell death
· 231: Smac/DIABLO
· 241: in H-ras-transformed human mammary epithelial cells
· Be consistent in typing proteins or genes (eg. BAX or Bax; ICAM1 or ICAM-1; p38 or P38), chosing the more appropriate form
· Cyclin-dependent kinases should be abbreviated as CDKs not Cdks
· 268: G0/G1
· 276: epithelial-mesenchymal transition (EMT)
· 274-286: Add some reference
· 291: remove a )
· 299: glucose uptake
· 299: reported that elevated
· 302: which is a transcription factor
· 316: Phloretin as a potential cancer immunotherapy agent
· 319: programmed cell death 1 (PD-1)
· 320: PD-L1
· 322: progress has been made
· 336-337: Add some reference
· 336-337: this is not only mechanism as the Author seems to suggest
· 338-9: P-glycoprotein
· 365: to prevent or delay
Reviewer 2 Report
This article provides a good review for the anti-cancer and anti-inflammatory effects of phloretin, a natural polyphenol obtained from apple. It contains content with reference value and its organizational structure is appropriate. Therefore, I recommend to accept it be published in this journal.
Specific comment:
1. The References section does not quite match the format of this journal. Please check carefully.
Author Response
Journal Molecules (ISSN 1420-3049)
Manuscript ID: Molecules-423825
Title : Biochemical basis of anti-cancer effects of Phloretin – A Natural Dihydrochalcone
Reviewer #2
Comments and Suggestions for Authors
This article provides a good review for the anti-cancer and anti-inflammatory effects of phloretin, a natural polyphenol obtained from apple. It contains content with reference value and its organizational structure is appropriate. Therefore, I recommend to accept it be published in this journal.
Specific comment:
1. The References section does not quite match the format of this journal. Please check carefully.
Response
Thank you for your good comment. References section has been modified to fit molecules. EndNote has been modified to use.
Round 2
Reviewer 1 Report
The Author significantly improved the manuscript as requested, thus I recommend accepting it after the following small corrections:
· Line 134: it should be “pharmacophore for the radical scavenging and lipid peroxidation activities”
o By mistake in the previous comment I wrote line 132 instead of 134. It was correct as previously written (“2,6-dihydroxyacetone moiety as the 130 pharmacophore [29]”).
· Line 341: added reference should be numbered
· Line 380: delete: “(Hannah Deußer et al., Biotechnol. J. 2013, 8, 363–370)”
Author Response
Journal Molecules (ISSN 1420-3049)
Manuscript ID: Molecules-423825
Title : Biochemical basis of anti-cancer effects of Phloretin – A Natural Dihydrochalcone
Reviewer #1
The Author significantly improved the manuscript as requested, thus I recommend accepting it after the following small corrections:
Comment 1
· Line 134: it should be “pharmacophore for the radical scavenging and lipid peroxidation activities”
o By mistake in the previous comment I wrote line 132 instead of 134. It was correct as previously written (“2,6-dihydroxyacetone moiety as the 130 pharmacophore [29]”).
· Line 341: added reference should be numbered
· Line 380: delete: “(Hannah Deußer et al., Biotechnol. J. 2013, 8, 363–370)”
Response 1
Thank you for your comments. And I have finished editing the comments. The modifications were highlighted in sky blue in manuscript.